# Prevalence of depressive symptoms among secondary school adolescents in Dodoma and Pwani, Tanzania

Divine Patrick Mwaluli[1]*, Amani Angumbwike Mwakalapuka[1], Joshua Joel Matiku[2], Jamal Jumanne Athuman[1]

1 Department of Educational Psychology and Counselling, School of Education, Sokoine University of Agriculture, Morogoro, Tanzania, 2 Department of Education, Mwalimu Nyerere University of Agriculture and Technology, Mara, Tanzania

* mwima.joshua@gmail.com

## Abstract

Adolescent depression is a growing public health concern, particularly in low- and middle-income countries where mental health resources are limited. This study aimed to assess the prevalence of depressive symptoms among secondary school adolescents in the Dodoma and Pwani Regions of Tanzania and to explore associations with key demographic factors. A cross-sectional survey was conducted among 1009 secondary school adolescents in the Dodoma and Pwani regions of Tanzania. The Swahili version of the Patient Health Questionnaire-9 (PHQ-9) was used to assess depressive symptoms. Descriptive statistics were used to determine prevalence, while associations between depression scores and demographic variables were analysed using Chi-square, one-way ANOVA, Mann-Whitney U, and Kruskal-Wallis H tests, with the magnitude of association reported as Cramér's V for the Chi-square test. Overall, 62% of adolescents exhibited depressive symptoms ranging from mild to severe, with 37% mild, 17% moderate, 6% moderately severe, and 2% severe. Depressive symptoms were significantly associated with age, region, gender, and parental living arrangement ($p < 0.05$). A higher prevalence was observed among females, adolescents aged 14–17 years, residents of Dodoma, and those living with a single parent. Effect size analysis indicated a strong association with region and a moderate association with parental living arrangement, while gender and age showed smaller but meaningful associations. Depression among Tanzanian adolescents in Dodoma and Pwani is prevalent and significantly associated with key demographic and psychosocial factors.

**Data availability statement:** The minimal anonymized dataset supporting the findings of this study is publicly available in the Zenodo repository and can be accessed at: https://doi.org/10.5281/zenodo.20201917. The dataset was anonymized prior to public sharing in accordance with the ethical requirements approved by the SUA University Research and Publications Committee (URPC).

**Funding:** The authors received no specific funding for this work.

**Competing interests:** The authors have declared that no competing interests exist.

# 1 Introduction

## 1.1 Background to the problem

Depressive symptoms among adolescents are widely recognised as a major global public health concern because they affect emotional well-being, academic functioning, and social development, and they are also associated with increased risk of suicide and other adverse life outcomes [1,2]. Globally, a substantial proportion of adolescents aged 10–19 years are at risk of developing clinical depression or elevated depressive symptoms, indicating that mental health problems during adolescence are both common and consequential [1]. Early-onset depressive symptoms are particularly concerning because they may persist into adulthood and contribute to chronic mental health problems, substance abuse, and impaired life opportunities [3,4].

In sub-Saharan Africa, adolescent mental health has received limited attention despite the region's rapidly growing youth population and the multiple socioeconomic stressors affecting young people. Available evidence indicates that depressive symptoms are prevalent among adolescents in the region, with rates ranging from about 20% to over 40% in some settings [5,6]. These patterns are commonly linked to poverty, inadequate access to mental health services, violence exposure, and the burden of chronic illness, including HIV/AIDS [6]. In Tanzania, adolescents constitute a large proportion of the population, yet research and services addressing adolescent mental health remain limited, particularly in school contexts where young people spend much of their time [7,8].

Although previous studies have contributed to understanding adolescent mental health, much of the existing evidence in Tanzania has focused on selected vulnerable groups rather than the broader population of in-school adolescents. Furthermore, factors such as age, gender, and family living arrangements, which have been associated with depressive symptoms in the wider literature, remain insufficiently examined in the Tanzanian school context [5,9,10]. Therefore, this study was designed to assess the prevalence of depressive symptoms among adolescents in secondary schools in Dodoma and Pwani, Tanzania.

## 1.2 Statement of the problem

Adolescent depressive symptoms are an emerging public health concern in Tanzania, yet empirical evidence remains fragmented and concentrated in specific high-risk populations. Existing studies have reported a prevalence of 29.9% among adolescents in the Southern Highlands, including a substantial proportion of adolescents living with HIV [6]. Other studies have shown 36% prevalence among out-of-school adolescent girls and young women in Shinyanga [11], while research in Moshi documented a 15% prevalence of psychosocial and emotional challenges linked to puberty and menstruation among school-age girls [12]. In addition, hospital records from Mirembe National Mental Health Hospital for the period 2020–2023 indicated a growing number of depression cases, with higher burden reported in regions such as Dodoma, Morogoro, and Singida, compared to

lower levels in areas such as Pwani. These findings suggest that depressive symptoms are becoming an important mental health issue in Tanzania, but the available evidence is still scattered and unevenly distributed across population groups.

Despite these concerning patterns, most Tanzanian studies have focused on adolescents living with HIV, out-of-school girls, or those facing reproductive health challenges, leaving limited evidence on in-school adolescents, who make up a substantial proportion of the country's adolescent population [6,11,12]. This is a critical gap because in-school adolescents also experience academic pressure, social challenges, emotional stress, and developmental transitions that may increase vulnerability to depressive symptoms. Moreover, important correlates such as age, gender, and family living arrangements have not been adequately examined in the Tanzanian context, even though international literature consistently links older age, female gender, and non-intact family structure with elevated risk of depression [5,9,10].

The absence of comprehensive evidence on depressive symptoms among in-school adolescents limits the ability of policymakers, educators, and health professionals to design effective school-based mental health interventions and contextually relevant prevention strategies. Therefore, this study addresses this gap by examining the prevalence of depressive symptoms among secondary school adolescents in Dodoma and Pwani, Tanzania, and by assessing their association with age, gender, and parental living arrangements.

## 1.3  General objective of the study

To assess the prevalence of depressive symptoms among secondary school adolescents in selected regions of Tanzania and to examine their association with selected demographic factors, including age, gender, and Parental living arrangement.

### 1.3.1  Specific objectives.  This study intends:

- To assess the prevalence of depressive symptoms among secondary school adolescents in Dodoma and Pwani, Tanzania.

- To examine the association between depressive symptoms and demographic variables, including region, age, gender, and parental living arrangement, among secondary school adolescents.

## 1.4  Significance of the study

This study examines the mental health of secondary school adolescents in Tanzania, emphasising the prevalence and symptoms of depression and their variation by age, gender, region, and parental living arrangements. The findings offer valuable insights into demographic and socioeconomic factors linked to adolescent depression. By identifying higher-risk groups, such as older adolescents, females, and those living with a single parent, the study highlights the need for more targeted and inclusive mental health programs.

It emphasises the need to integrate mental health screening and support services into schools to facilitate early identification and timely intervention. Since schools are often the first environment where emotional and behavioural problems become evident, they serve as ideal settings for preventative and supportive actions.

In addition to its educational significance, the study provides practical guidance for policymakers, educators, and mental health professionals on developing evidence-based strategies to promote adolescent well-being. These initiatives can enhance students' emotional resilience, increase their academic engagement, and promote their long-term psychological well-being.

Ultimately, the findings support Tanzania's broader national objectives for youth health and development by emphasising the importance of evidence-based planning and fair resource allocation for adolescent mental health services.

## 2 Materials and methods

### 2.1 Study design

This study employed a **cross-sectional survey design** to assess the prevalence of depressive symptoms and their association with selected demographic factors among secondary school adolescents in Dodoma and Pwani, Tanzania. The cross-sectional design was considered appropriate because it allows for the collection of data at a single point in time from a defined population, enabling the estimation of prevalence and the examination of relationships between variables. This design is widely used in epidemiological and public health research due to its efficiency in capturing population-level patterns and associations within a relatively short period.

### 2.2 Study area and justification

This study was conducted in the Dodoma and Pwani regions, selected based on Mirembe National Mental Health Hospital records (2020–2023), which showed high depression rates in Dodoma and lower rates in Pwani. Although these data encompass all ages, they provide valuable background information on regions with distinct mental health profiles.

Dodoma, the centrally located capital, has experienced rapid urban growth and socioeconomic changes, which may have increased psychosocial stressors. Meanwhile, Pwani remains predominantly rural and peri-urban, with slower urbanisation and different social and environmental factors affecting mental health. Including these regions enabled the study to collect adolescent depression data from diverse settings, enhancing the variety and context of the findings. This supports the study's aim to evaluate depressive symptoms and their relation to demographic factors across various Tanzanian settings.

### 2.3 Participants and sampling procedures

The study focused on secondary school adolescents aged 11–19 years, enrolled in public schools in Dodoma and Pwani regions. These age groups were chosen because of their increased vulnerability to depression during adolescence. Data collection and participant recruitment took place between April 5, 2025, and May 30, 2025**.**

A simple random sampling method was employed. From a population of 29,719 students (10,258 in Pwani and 19,461 in Dodoma), 20 secondary schools were selected by lottery. Numbered slips representing each school listed in the region were drawn to ensure randomness. The sample included 10 schools from each region. Within each selected school, an equal number of participants were randomly selected from Form I and Form IV classes (typically aged 11–19), ensuring a balance of younger and older adolescents. This stratification by class year helped capture a broad age range. Parental consent and student assent were obtained before participation.

### 2.4 Data collection tools and procedures

Data were collected through face-to-face interviews using a structured questionnaire administered via KoBoToolbox on tablets. The questionnaire included the Swahili version of the PHQ-9 to screen for depressive symptoms, as well as additional items capturing demographic information (age, gender, school, class year, and living arrangement). The PHQ-9 has been validated in Swahili and has been used in adolescent populations in East Africa, making it an appropriate tool for this study. Scores on the PHQ-9 range from 0 to 27, and for analysis, standard severity cut-offs were used: 0–4 (no depression), 5–9 (mild depression), 10–14 (moderate depression), 15–19 (moderately severe depression), and 20–27 (severe depression).

Before the main data collection, a pilot study was conducted with 150 students from a school not included in the final sample. The aim was to assess the clarity, cultural appropriateness, and technical functioning of the Swahili PHQ-9, as well as to evaluate the feasibility of using KoBoToolbox, a mobile-based digital data collection platform. The pilot indicated that all items in the PHQ-9 were clear and culturally suitable, requiring no modifications to the tool itself. However, minor

changes were made to the questionnaire format by adding demographic questions such as age, gender, location, and parental living arrangements to improve contextual analysis. These adjustments did not compromise the validity or reliability of the PHQ-9 but did make the data more comprehensive and improve data collection efficiency. KoBoToolbox was selected for the main study because of its proven reliability in psychosocial research involving humans, including evidence from a large-scale child maltreatment survey in Vietnam, which demonstrated the platform's ability to build complex questionnaires efficiently, include validation to reduce data entry errors, provide a user-friendly interface, and operate offline in areas with limited connectivity [13]. Data collection was conducted in the selected schools by the principal investigator, with support from two trained research assistants, enabling efficient, secure, and accurate real-time data capture in the field.

## 2.5 Data analysis

Completed questionnaires were downloaded from KoBoToolbox and analysed using statistical software (SPSS Version 27). Descriptive statistics (frequencies and percentages) summarised the prevalence of depressive symptoms overall and by severity category. Associations between depression (both as continuous scores and categorical severity levels) and demographic factors were assessed as outlined: Chi-square tests for categorical associations (with Cramér's V or Phi for effect sizes), and Mann-Whitney U or Kruskal-Wallis H for score comparisons between groups (with effect size r). The significance level was set at $p < 0.05$ for hypothesis tests (with Bonferroni correction applied for multiple comparisons in post hoc tests). Results are presented in tables and figures, accompanied by narrative explanations.

## 2.6 Ethical considerations and approval

The study adhered to the Sokoine University of Agriculture (SUA) Code of Conduct for Research Ethics, with ethical approval obtained from the University Research and Publication Committee (URPC). Since the study involved behavioural and psychosocial assessments rather than clinical interventions, approval from the National Health Research Ethics Review Committee (NHRERC) under the National Institute for Medical Research (NIMR) was not necessary. Administrative permissions were obtained from the Regional and District Education Offices and from the heads of participating schools.

In accordance with the granted ethical clearance, institutional permission was obtained from relevant educational authorities and school heads prior to data collection. For participants aged below 18 years, written informed consent was obtained from parents or guardians through school administration, with school heads facilitating the consent process. All adolescents below 18 years provided written informed assent before participation. Participants aged 18 years and above provided their own written informed consent. Participation was voluntary, confidentiality was maintained, and participants were informed of their right to withdraw at any time. Adolescents who exhibited signs of psychological distress were referred to a school counsellor or an available mental health professional.

## 3 Results

### 3.1 Prevalence of depressive symptoms among secondary school adolescents in selected regions

The findings show that 38% of respondents reported no depressive symptoms. Conversely, 37% exhibited symptoms of mild depression, while 17% were classified as experiencing moderate depression. Additionally, 6% reported moderate to severe depression, and 2% were identified with severe depression. Overall, these results suggest that 62% of the sample experienced some level of depressive symptoms, from mild to severe. This indicates that most participants are affected by varying degrees of depression, highlighting a concerning trend in overall mental health within the study group. Below is a bar chart presenting the distribution of depression levels among participants (Fig 1).

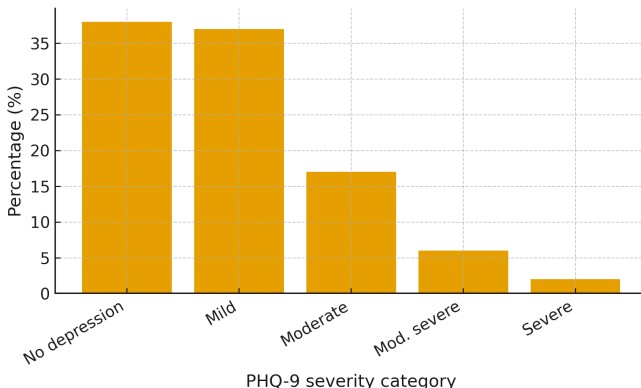

**Fig 1. Distribution of depression levels among participants.**

Age: There was an observable increase in depression prevalence with age. Among adolescents aged 11–13, 55% reported depressive symptoms (PHQ-9 ≥ 5). This proportion rose to 63% for those aged 14–16 and further to 67% for those aged 17–19. Similarly, the proportion with moderate-to-severe depression (PHQ-9 ≥ 10) was 20% for 11–13 years, 25% for 14–16 years, and 28% for 17–19 years. This trend suggests mid-to-late adolescence (approximately ages 14–19) is associated with higher rates of depressive symptoms.

Gender: Female adolescents exhibited a higher prevalence of depressive symptoms compared to males. Approximately 66% of female students had PHQ-9 scores of 5 or higher, compared to 56% of male students. For moderate-to-severe depression, the figures were 27% for females and 21% for males. This indicates that girls in the sample were more likely to experience depressive symptoms, aligning with global patterns of higher depression rates in adolescent females.

Region: Adolescents in Dodoma showed a higher prevalence of depressive symptoms than those in Pwani. In Dodoma, 64% had PHQ-9 ≥ 5 and 26% had PHQ-9 ≥ 10, whereas in Pwani 59% had PHQ-9 ≥ 5 and 23% had PHQ-9 ≥ 10. The urbanized and rapidly changing context of Dodoma may contribute to this difference, compared to the more rural Pwani.

Parental Living Arrangement: Adolescents living with a single parent exhibited the highest prevalence of depressive symptoms: 69% had PHQ-9 scores of 5 or higher, and 30% had PHQ-9 scores of 10 or higher. In contrast, those living with both parents showed 59% with a PHQ-9 score of 5 or more and 23% with a PHQ-9 score of 10 or more. Adolescents living with other guardians had intermediate rates (62% with PHQ-9 ≥ 5; 25% with PHQ-9 ≥ 10). These descriptive patterns suggest that not living with both parents is linked to higher rates of adolescent depression. Table 1 displays the distribution of adolescents across five depression severity categories (no depression, mild, moderate, moderate-to-severe, and severe) within each demographic variable (region, gender, age group, and parental living arrangement). Values are presented as frequencies and percentages (%).

### 3.2 Associations and effect sizes between depressive symptoms and demographic characteristics

Chi-square tests indicated significant associations for region (p < 0.001), age group (p < 0.001), and parental living arrangment (p = 0.023), while gender showed a borderline significant link (p = 0.056). Cramér's V showed that the region had the strongest relationship with depressive symptoms (V = 0.447, large effect size), with adolescents in Dodoma reporting more moderate to severe symptoms than those in Pwani. The age group also displayed a strong association (V = 0.397), with late adolescents (17–19) being more likely to experience higher levels of depression than their younger peers. parental living arrangement (V = 0.271) and gender (V = 0.255) demonstrated moderate effects, indicating that adolescents living with a single parent or other guardians, as well as females, tended to report higher depression. Overall, region and age group were the most influential factors, followed by parental living arrangment and gender, highlighting the importance of

**Table 1. Prevalence of depressive symptom across demographic subgroups.**

| Variable | Category | No Depression | Mild Depression | Moderate Depression | Moderate to Severe | Severe Depression |
|---|---|---|---|---|---|---|
| Region | Dodoma | 163 (42%) | 186 (49%) | 112 (66%) | 37 (66%) | 12 (67%) |
| Region | Pwani | 225 (58%) | 191 (51%) | 58 (34%) | 19 (34%) | 6 (33%) |
| Gender | Male | 205 (53%) | 172 (46%) | 76 (45%) | 23 (41%) | 5 (28%) |
| Gender | Female | 183 (47%) | 205 (54%) | 94 (55%) | 33 (59%) | 13 (72%) |
| Age Group | Early adolescence (11–13) | 53 (14%) | 32 (9%) | 12 (7%) | 5 (9%) | 1 (6%) |
| Age Group | Middle adolescence (14–16) | 258 (67%) | 237 (63%) | 99 (58%) | 26 (46%) | 10 (56%) |
| Age Group | Late adolescence (17–19) | 77 (20%) | 108 (29%) | 59 (35%) | 25 (45%) | 7 (39%) |
| parental living arrangement | Lived with other guardians | 136 (35%) | 109 (29%) | 59 (35%) | 14 (25%) | 4 (22%) |
| parental living arrangement | Lived with a single parent | 163 (42%) | 158 (42%) | 72 (42%) | 32 (57%) | 13 (72%) |
| parental living arrangement | Lived with both parents | 89 (23%) | 110 (29%) | 39 (23%) | 10 (18%) | 1 (6%) |

considering geographic, developmental, and family contexts when designing school-based mental health interventions. Table 2 shows the association between depressive symptoms and adolescents' background characteristics.

**3.2.1 Differences in the depressive symptoms between adolescents in the Dodoma and Pwani Regions.** The results of the Independent-Samples Mann-Whitney U Test show a statistically significant difference in depression scores (PHQ-9 total scores) between adolescents in Dodoma and Pwani regions. The analysis produced a Mann-Whitney U statistic of 104273.5 and a Wilcoxon W statistic of 229023.5. With a standardised test statistic of -4.976 and an asymptotic significance (two-sided test) of less than 0.001, the study confidently concludes that adolescents in one region have significantly different depression levels compared to those in the other. This highly significant finding suggests that the differences in depression scores across Dodoma and Pwani are not due to random chance but indicate meaningful disparities in depression experienced by adolescents in these two regions of Tanzania (Table 3).

To supplement the Mann-Whitney U Test results, a histogram (Fig 2) was created to visually compare the distribution of PHQ-9 depression scores between adolescents in Dodoma and Pwani regions. A visual comparison of PHQ-9 depression scores between adolescents in Dodoma and Pwani reveals an apparent disparity. Dodoma has a higher mean rank (550.04) than Pwani (458.96), indicating more severe depressive symptoms. The 91-point difference highlights a

**Table 2. Prevalence of depressive symptoms and associated test statistics across demographic characteristics.**

| Variables | | PHQ9 | | | | | Test statistics | | |
|---|---|---|---|---|---|---|---|---|---|
| | | No depression | Mild depression | Moderate depression | Moderate to severe depression | Severe depression | df (X²) | P – value | Cramer's V |
| Regions | Dodoma | 163 (42%) | 186 (49%) | 112 (66%) | 37 (66%) | 12 (67%) | 4(34.79) | <0.001*** | 0.447 |
| | Pwani | 225 (58%) | 191 (51%) | 58 (34%) | 19 (34%) | 6 (33%) | | | |
| Gender | Male | 205 (53%) | 172 (46%) | 76 (45%) | 23 (41%) | 5 (28%) | 4(9.21) | 0.056* | 0.255 |
| | Female | 183 (47%) | 205 (54%) | 94 (55%) | 33 (59%) | 13 (72%) | | | |
| Age group | Early adolescence (11–13) | 53 (14%) | 32 (9%) | 12 (7%) | 5 (9%) | 1 (6%) | 8(30.28) | <0.001*** | 0.397 |
| | Middle adolescence (14–16) | 258 (67%) | 237 (63%) | 99 (58%) | 26 (46%) | 10 (56%) | | | |
| | Late adolescence (17–19) | 77 (20%) | 108 (29%) | 59 (35%) | 25 (45%) | 7 (39%) | | | |
| parental living arrangement | Lived with other guardians | 136 (35%) | 109 (29%) | 59 (35%) | 14 (25%) | 4 (22%) | 8(17.83) | 0.023** | 0.271 |
| | Lived with a single parent | 163 (42%) | 158 (42%) | 72 (42%) | 32 (57%) | 13 (72%) | | | |
| | Lived with both parents | 89 (23%) | 110 (29%) | 39 (23%) | 10 (18%) | 1 (6%) | | | |

**Table 3. Differences in the Depressive Symptoms Between Adolescents in Dodoma and Pwani.**

| Statistic | Value |
|---|---|
| Mann–Whitney U | 104273.5 |
| Wilcoxon W | 229023.5 |
| Test Statistic | 104273.5 |
| Standard Error | 4616.224 |
| Standardized Test Statistic (Z) | −4.976 |
| **p-value** | **< 0.001*** |

\*\*\* p < .01, \*\* p < .05, \* p < .1

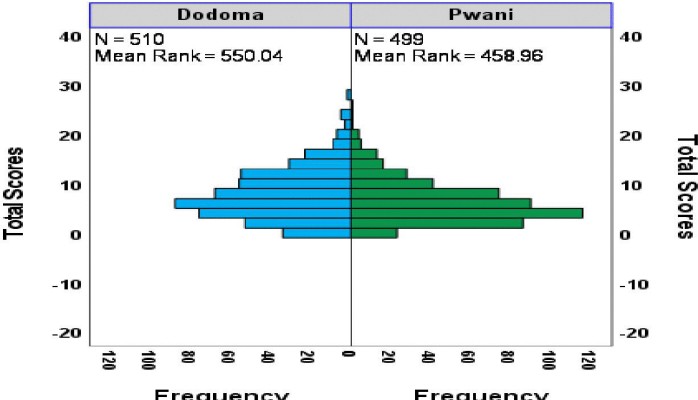

**Fig 2. Differences in distributions of PHQ-9 scores between Dodoma and Pwani.**

significant variation, with Dodoma's scores skewed higher and Pwani's clustered lower. This distinction is confirmed by a statistically significant Mann-Whitney U Test (Z = -4.976), emphasising a real and notable difference in depression levels between the two regions.

**3.2.2 Gender differences in depressive symptoms among secondary school adolescents.** The Independent-Samples Mann-Whitney U Test reveals a significant difference in PHQ-9 depression scores between male and female adolescents. With a U statistic of 138169.5, a Wilcoxon W of 277825.5, and a standardised test statistic of 2.426, the results indicate differing depression levels between genders. The p-value of 0.015 (p < 0.05) confirms statistical significance. This moderate yet meaningful effect suggests that gender influences depressive symptoms and highlights the need for further research into gender-specific factors. Below is Table 4, which presents the data.

Fig 3 clearly shows different distributions of PHQ-9 depression scores between female and male adolescents. Females (mean rank = 526.18, N = 528) report higher scores than males (mean rank = 481.75, N = 481), with a notable difference of 44.43 points. The histogram illustrates that females experience more frequent and severe symptoms, as their scores extend further into higher ranges, whereas male scores tend to cluster at lower levels. This visual pattern aligns with the Mann-Whitney U Test results (Z = 2.426, p = 0.015), confirming a statistically significant gender difference in depressive symptoms among Tanzanian adolescents.

**Table 4. Mann-Whitney U Test by gender.**

| Statistic | Value |
|---|---|
| Mann–Whitney U | 138169.5 |
| Wilcoxon W | 277825.5 |
| Test Statistic | 138169.5 |
| Standard Error | 4611.488 |
| Standardized Test Statistic (Z) | 2.426 |
| **p-value** | **0.015*** |

*** p < .01, ** p < .05, * p < .1

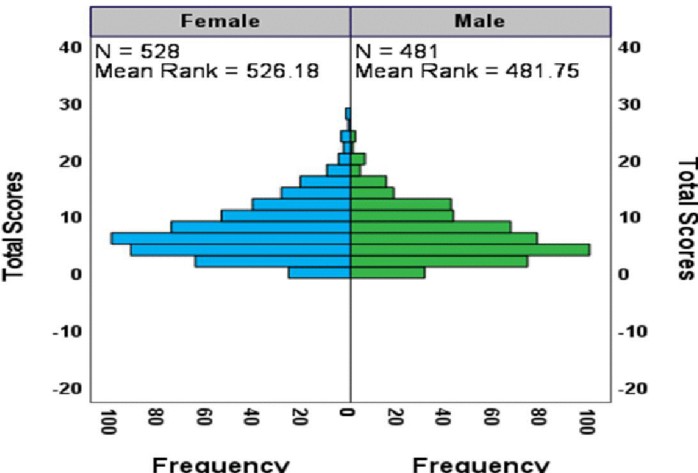

**Fig 3. Differences in distributions of PHQ-9 scores between female and male adolescents.**

**3.2.3 Differences in depressive symptoms among adolescents living in different parental arrangements.** The Kruskal-Wallis H test indicated a statistically significant difference in depressive symptoms (measured by PHQ-9 ranks) across various parental living arrangment groups ($\chi^2$ (2) = 1.936, p = 0.048), demonstrating that a student's household structure is meaningfully linked to their level of depression. Adolescents living with a single parent had the highest mean rank (518.36), suggesting more depressive symptoms, followed by those living with both parents (502.35), with the lowest rank among those living with other guardians (488.88). Post hoc pairwise comparisons further showed that adolescents living with a single parent were significantly more depressed than those living with other guardians (p = 0.036) and those living with both parents (p = 0.022). A marginally significant difference was also observed between adolescents living with other guardians and those living with both parents (p = 0.059). These findings consistently indicate that adolescents from single-parent households are more vulnerable to depression, highlighting the emotional and psychological challenges linked to limited parental support. The differences in ranks, especially the 29.48-point range, emphasise how family structure can influence adolescents' mental health, with implications for targeted school and community-based interventions in Tanzania. Tables 5 and 6 present these findings.

# 4 Discussion

The findings of this study reveal a substantial burden of depressive symptoms among secondary school adolescents in Tanzania, indicating that adolescent mental health remains a critical public health concern. The high prevalence observed

**Table 5. Depressive symptoms by parental status.**

| parental living arrangement | N | Mean Rank | χ² (df = 2) | p-value |
|---|---|---|---|---|
| Lived with other guardians | 322 | 488.88 | | |
| Lived with a single parent | 438 | 518.36 | **1.936** | **0.048 **** |
| Lived with both parents | 249 | 502.35 | | |

*** p<.01, ** p<.05, * p<.1

**Table 6. Pairwise comparisons of depressive symptoms by parental living arrangement.**

| Comparison | Mean Difference | p-value |
|---|---|---|
| Other guardians vs. Single parent | −1.38149 | 0.036** |
| Other guardians vs. Both parents | −0.54932 | 0.059* |
| Single parent vs. Both parents | 0.693674 | 0.022** |

*** p<.01, ** p<.05, * p<.1

is consistent with emerging evidence from sub-Saharan Africa, where elevated levels of depressive symptoms have been widely reported among adolescents [5,14]. Similar findings have been documented in East African contexts, where studies using symptom-based screening tools have identified a large proportion of adolescents experiencing mild to severe depressive symptoms [5,15]. This pattern may reflect the cumulative impact of contextual stressors such as economic hardship, family instability, academic pressure, and limited access to mental health services, which collectively increase adolescents' vulnerability to emotional distress.

The observed increase in depressive symptoms with age, particularly during mid-to-late adolescence, aligns with developmental and epidemiological evidence identifying this period as a critical stage for the onset of mental health problems [2, 4]. During this phase, adolescents experience significant biological, cognitive, and social transitions, including identity formation, increased academic demands, and evolving peer relationships, all of which may heighten emotional sensitivity and stress. Similar age-related trends have been reported across low- and middle-income countries, where depressive symptoms tend to intensify during secondary school years [16]. These findings support developmental psychopathology perspectives that conceptualise adolescence as a key window of vulnerability for affective disorders.

Regional differences in depressive symptoms further highlight the influence of contextual and environmental factors on adolescent mental health. The higher levels observed in more urbanised settings, such as Dodoma, suggest that urbanisation-related stressors, including social inequality, academic competition, and weakening social cohesion, may contribute to increased psychological distress. Comparable urban–rural disparities have been reported in other African settings, including Nigeria, where adolescents in urban environments often report higher levels of depression [16]. However, these findings also underscore the importance of local context, as patterns may vary depending on specific socio-economic and cultural conditions.

Family living arrangement was also significantly associated with depressive symptoms, reinforcing the importance of family context in adolescent mental health. Consistent with previous studies, adolescents from single-parent households exhibited higher levels of depressive symptoms, likely due to economic strain, reduced emotional support, and increased responsibilities [17,18]. Evidence from African contexts similarly indicates that adolescents lacking stable parental support are more vulnerable to psychological distress [11]. However, the present findings also demonstrate that family structure alone does not determine mental health outcomes. Many adolescents in two-parent households also reported significant

depressive symptoms, suggesting that the quality of family relationships, levels of conflict, and availability of emotional support may be more influential than family structure itself. This highlights the need to move beyond simplistic categorizations and focus on family functioning as a key determinant of adolescent well-being.

Gender differences observed in this study are consistent with global and regional literature indicating higher vulnerability to depressive symptoms among female adolescents [19,20]. Meta-analytic evidence suggests that adolescent girls are approximately twice as likely to experience depressive symptoms as boys [21], a pattern often attributed to a combination of biological, psychological, and sociocultural factors. Hormonal changes during puberty may increase emotional sensitivity, while gendered social expectations and exposure to stressors such as harassment and inequality further contribute to vulnerability. At the same time, the relatively modest effect size observed in this study suggests that depressive symptoms are also prevalent among boys, underscoring the need for inclusive mental health interventions that address the needs of all adolescents.

The findings of this study have important practical and policy implications. The high prevalence of depressive symptoms underscores the urgent need for school-based mental health interventions in Tanzania. Schools provide a strategic platform for early identification and support, particularly through routine screening using validated tools such as the PHQ-9. Integrating mental health services into school health programmes, training teachers to recognise signs of distress, and establishing referral pathways can facilitate timely intervention. Additionally, targeted interventions should focus on vulnerable groups, including older adolescents, girls, and those from disadvantaged family backgrounds.

From a theoretical perspective, the findings support key assumptions of Social Cognitive Theory, which emphasises the interaction between personal, behavioural, and environmental factors in shaping individual outcomes. The observed influence of age, gender, and family context highlights the complex interplay between individual characteristics and environmental conditions in determining adolescent mental health. Furthermore, the results are consistent with developmental psychopathology frameworks, which identify adolescence as a critical period for the emergence of emotional disorders due to rapid developmental changes [22].

## 4.1 Strengths and limitations

Strengths of this study include the large, randomly selected sample, which enhances the generalisability to secondary school adolescents in the two regions. The use of a validated screening tool (PHQ-9) ensures that clinically relevant symptoms are captured. Also, the study employed rigorous statistical analyses, including effect size measures, to complement p-values and provide a deeper understanding of practical significance.

However, several limitations must be acknowledged. First, the cross-sectional design limits causal inferences; one cannot conclude that factors like single parenthood cause depression, only that they co-occur. Longitudinal studies would be needed to parse directionality (for instance, whether adolescent depression might strain family relationships, as well as vice versa). Second, data were based on self-reported symptoms without clinical interviews, which may overestimate "depression" prevalence since not all who score high on PHQ-9 would meet diagnostic criteria. Nonetheless, the PHQ-9's strong validation in similar contexts supports its utility as a screening measure.

Third, while the study examined several demographic factors, it did not assess many other potentially important variables, such as experiences of trauma, bullying, substance use, physical health conditions, family conflict, or school-related factors. Depression is multiply determined, and the variables measured explained only a small portion of the variance in depression scores. Future research incorporating a broader range of risk and protective factors would provide a more complete picture.

Finally, the study cannot rule out the possibility that some regional or demographic differences reflect variations in willingness to report symptoms rather than actual differences in depression. However, the consistency of gender findings with international research and the use of a validated instrument provide some confidence in the validity of our results.

PLOS Mental Health

## 4.2 Unexpected results

Several unexpected findings emerged from the study. First, while the Chi-square analysis did not reveal significant gender differences in depressive symptoms, the Mann-Whitney U test did, suggesting that nonparametric rank-based methods may be more sensitive in detecting subtle differences in ordinal data, such as depression scores. Second, contrary to expectations, adolescents in the more urbanised Dodoma region reported higher depression levels than those in the Pwani region. This pattern may reflect the influence of urban-related stressors, such as academic competition, increased cost of living, and reduced social cohesion. Finally, the presence of moderately severe to severe depression among a notable proportion of in-school adolescents challenges the common assumption that school attendance serves as a protective factor against mental health problems, indicating that other contextual and psychosocial factors may override the potential benefits of being in school, as attendance itself may not be solely protective.

# 5 Conclusion

This study highlights depression as a significant mental health concern among secondary school adolescents in Tanzania, with 62% of participants reporting symptoms ranging from mild to severe. Mild depression was identified as the most common form, emphasising the need for early detection and intervention before symptoms worsen. The findings reveal a clear pattern based on age, with adolescents aged 14–17 being the most affected, as shown in Table 1. This suggests that mid-adolescence is a critical period when depressive symptoms are more likely to develop. While gender differences were not consistent across all analyses, female students consistently showed higher depression scores, indicating a need for gender-sensitive mental health programmes. Refer to Table 1 and Fig 2.

Regional differences were also observed, as shown in Table 3 and Fig 1; for instance, adolescents in Dodoma were more prone to depression than those in Pwani, likely due to socioeconomic challenges and unequal access to mental health services. Additionally, adolescents living with single parents were more likely to experience severe depression, highlighting the importance of family involvement and social support networks in promoting emotional well-being.

The effect size analysis (refer to Table 2) revealed that geography and parental living arrangements had moderate correlations with depressive symptoms, whereas gender had a smaller but still significant impact. Collectively, these data demonstrate how demographic, social, and environmental factors interact to influence adolescent mental health. To effectively reduce the burden of depression among Tanzanian adolescents, the study advocates for integrated, context-specific, and demographically tailored strategies, including school-based mental health screening, family and community support, and equitable distribution of mental health resources across regions.

## 5.1 Recommendations

Based on the findings of this study, the following recommendations are made:

- Implement school-based mental health screening programs to identify adolescents at risk of depression and provide early intervention.

- Develop targeted interventions for high-risk groups, including female adolescents, older adolescents, and those living in single-parent households.

- Integrate mental health education into the secondary school curriculum to increase awareness and reduce stigma related to mental health issues.

- Strengthen mental health services in schools by training teachers and school counsellors to recognise and respond to signs of depression.

- Conduct further research to explore the underlying factors contributing to regional differences in depression prevalence and to identify effective interventions for reducing the burden of adolescent depression in Tanzania.

## Author contributions

**Conceptualization:** Divine Patrick Mwaluli.

**Data curation:** Divine Patrick Mwaluli, Amani Angumbwike Mwakalapuka.

**Formal analysis:** Divine Patrick Mwaluli.

**Investigation:** Divine Patrick Mwaluli.

**Methodology:** Divine Patrick Mwaluli.

**Project administration:** Divine Patrick Mwaluli.

**Resources:** Divine Patrick Mwaluli.

**Software:** Divine Patrick Mwaluli.

**Supervision:** Amani Angumbwike Mwakalapuka, Joshua Joel Matiku, Jamal Jumanne Athuman.

**Validation:** Divine Patrick Mwaluli.

**Visualization:** Divine Patrick Mwaluli.

**Writing – original draft:** Divine Patrick Mwaluli.

**Writing – review & editing:** Divine Patrick Mwaluli, Joshua Joel Matiku.

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
