## [Decision Letter · Decision Letter 0]

6 Apr 2026

PMEN-D-25-00519

Divine Patrick Mwaluli

Symptoms and Prevalence of Depression Among Adolescents in Dodoma and Pwani Secondary Schools in Tanzania

PLOS Mental Health

Dear Dr. mwaluli,

Thank you for submitting your manuscript to PLOS Mental Health. I am sorry for the delay in reaching a decision - this was due to difficulties securing reviewers. After careful consideration of the reviewer reports, which we have now received, we feel that your paper has merit but does not fully meet PLOS Mental Health’s publication criteria as it currently stands. Therefore, we invite you to submit a revised version of the manuscript that addresses the points raised during the review process.

Please address all of the comments made, which you can find at the end of this email and in the attachment.

We look forward to receiving your revised manuscript.

Kind regards,

Karli Montague-Cardoso

Staff Editor

PLOS Mental Health

Journal Requirements:

1. During your revisions, please confirm whether the wording in the title is correct and update it in the manuscript file and online submission information if needed. Specifically, the corresponding author's name mentioned in the study title.

2. In the online submission form, you indicated that your data will be submitted to a repository upon acceptance. We strongly recommend all authors deposit their data before acceptance, as the process can be lengthy and hold up publication timelines. Please note that, though access restrictions are acceptable now, your entire minimal dataset will need to be made freely accessible if your manuscript is accepted for publication. This policy applies to all data except where public deposition would breach compliance with the protocol approved by your research ethics board. If you are unable to adhere to our open data policy, please kindly revise your statement to explain your reasoning and we will seek the editor's input on an exemption.

3. Please ensure that the Title in your manuscript and the Title in your online submission form are the same.

Additional Editor Comments (if provided):

Reviewers' comments:

Reviewer's Responses to Questions

**Comments to the Author**

1. Does this manuscript meet PLOS Mental Health’s publication criteria? Is the manuscript technically sound, and do the data support the conclusions? The manuscript must describe methodologically and ethically rigorous research with conclusions that are appropriately drawn based on the data presented.

Reviewer #1: Yes

Reviewer #2: Yes

2. Has the statistical analysis been performed appropriately and rigorously?

Reviewer #1: Yes

Reviewer #2: Yes

3. Have the authors made all data underlying the findings in their manuscript fully available (please refer to the Data Availability Statement at the start of the manuscript PDF file)?

Reviewer #1: Yes

Reviewer #2: Yes

4. Is the manuscript presented in an intelligible fashion and written in standard English?

Reviewer #1: Yes

Reviewer #2: Yes

5. Review Comments to the Author

Reviewer #1: The manuscript is well written, it highlights the major global health concern on mental health issues especially among young generation. The tools for data collection is the up to date and relevant to the study objectives.

In the ethical they should highlights which age category assent was sought and which age category informed consent was signed.

Reviewer #2: Overall comments

• The study is scientifically valuable and publishable

• Large sample size was used(n=1009)

• There was use of validated tool (PHQ-9 Swahili)

• Inclusion of effect sizes was a very good practise

• Strong public health relevance in Tanzania

• Clear policy and school-based implications on the recommendation

• Transparency on the methodology section.

Specific areas that need revision

Overall- grammatical checks should be performed throughout the document.

Abstract

Interchange of phrases “prevalence of depression” versus “prevalence of depressive symptoms”- based on the context of the tools used for screening depression “PHQ-9” and it is not a diagnostic tool for depression hence the author should use the term prevalence of depressive symptoms instead of prevalence of depression throughout the manuscript.

Title

Title says “Symptoms and Prevalence of Depression Among Adolescents in Dodoma and Pwani Secondary Schools in Tanzania” I think this should also be paraphrased because the study will focus on prevalence of depressive symptoms

Introduction

Instead of using “[4] found that 29.9%” use the name of author/place/type of study E.g , A case control study done in the southern highlands of Tanzania found that 29.9%.....

Introduction and Problem statement

There is some information in the introduction explaining the problem of mental health among adolescents in Tanzania which should actually be in the problem statement section. Some data repeatedly appear on both introduction and problem statement. Kindly restructure this.

Rationale- Well written

Methodology

Study design- on this section just explain about the study design, information about data collection tool used and data analysis should be removed from study design section and kept in the respective sections under methodology.

Results

Paragraph 1- “non-clinical mental health status” what does this mean?

Figures are missing

“Table 1 shows the association between depression symptoms and adolescents’ background characteristics.”- Ideally this is table 2 and not table 1, kindly correct this in the manuscript.

Discussion

Comparison studies section should be included within the discussion.

Avoid repetition of results in the discussion section

References

Please ensure consistent formatting (APA/Vancouver style)

6. PLOS authors have the option to publish the peer review history of their article (what does this mean?). If published, this will include your full peer review and any attached files.

**Do you want your identity to be public for this peer review?** For information about this choice, including consent withdrawal, please see our Privacy Policy.

Reviewer #1: No

Reviewer #2: No

Figure Resubmissions:

---

## [Decision Letter · Decision Letter 1]

12 May 2026

Prevalence of Depressive Symptoms Among Secondary School Adolescents in Dodoma and Pwani, Tanzania

PMEN-D-25-00519R1

Dear student mwaluli,

We are pleased to inform you that your manuscript 'Prevalence of Depressive Symptoms Among Secondary School Adolescents in Dodoma and Pwani, Tanzania' has been provisionally accepted for publication in PLOS Mental Health.

Best regards,

Karli Montague-Cardoso

Staff Editor

PLOS Mental Health

Reviewer Comments (if any, and for reference):

Reviewer's Responses to Questions

**Comments to the Author**

1. If the authors have adequately addressed your comments raised in a previous round of review and you feel that this manuscript is now acceptable for publication, you may indicate that here to bypass the “Comments to the Author” section, enter your conflict of interest statement in the “Confidential to Editor” section, and submit your "Accept" recommendation.

Reviewer #2: All comments have been addressed

2. Does this manuscript meet PLOS Mental Health’s publication criteria? Is the manuscript technically sound, and do the data support the conclusions? The manuscript must describe methodologically and ethically rigorous research with conclusions that are appropriately drawn based on the data presented.

Reviewer #2: Yes

3. Has the statistical analysis been performed appropriately and rigorously?

Reviewer #2: Yes

4. Have the authors made all data underlying the findings in their manuscript fully available (please refer to the Data Availability Statement at the start of the manuscript PDF file)?

Reviewer #2: Yes

5. Is the manuscript presented in an intelligible fashion and written in standard English?

Reviewer #2: Yes

6. Review Comments to the Author

Reviewer #2: i agree that the manuscript should go forward for publication. The author needs to do final grammar corrections on the work, otherwise its good for publication

7. PLOS authors have the option to publish the peer review history of their article (what does this mean?). If published, this will include your full peer review and any attached files.

**Do you want your identity to be public for this peer review?** For information about this choice, including consent withdrawal, please see our Privacy Policy.

Reviewer #2: No
